civil engineering/environmental engineering

urban resource systems, resource efficiency, random graphs, circular economy, mathematical modelling

**Author for correspondence:**
Hadi Arbabi
e-mail: h.arbabi@sheffield.ac.uk

# On the use of random graphs in analysing resource utilization in urban systems

Hadi Arbabi[1], Giuliano Punzo[2], Gregory Meyers[1], Ling Min Tan[1], Qianqian Li[1], Danielle Densley Tingley[1] and Martin Mayfield[1]

[1]Department of Civil and Structural Engineering, and [2]Department of Automatic Control and Systems Engineering, University of Sheffield, Sheffield S1 3JD, UK

  HA, 0000-0001-8518-9022; GM, 0000-0003-4157-3991

Urban resource models increasingly rely on implicit network formulations. Resource consumption behaviours documented in the existing empirical studies are ultimately by-products of the network abstractions underlying these models. Here, we present an analytical formulation and examination of a generic demand-driven network model that accounts for the effectiveness of resource utilization and its implications for policy levers in addressing resource management in cities. We establish simple limiting boundaries to systems' resource effectiveness. These limits are found not to be a function of system size and to be simply determined by the system's average ability to maintain resource quality through its transformation processes. We also show that resource utilization in itself does not enjoy considerable size efficiencies with larger and more diverse systems only offering increased chances of finding matching demand and supply between existing sectors in the system.

## 1. Introduction

Cities and their interconnected processes, be it those of their economic or industrial sectors, comprise complex interactions and their dynamics. In part due to urbanization and population growth, cities' growing demand has ever increasingly come to exceed planetary capacity. In this environmental context, and following analogies that are often drawn between thermodynamic systems and cities, cities can be thought of as open thermodynamic systems with their consumption mainly relying on incoming supplies of resources and energy from hinterlands outside their boundary. Such open systems are free to exchange matter and energy with their external environment to draw high-quality resources into the system for local transformation and consumption [1].

Thermodynamically speaking, the rate of utilization of resources in cities is then subject to the quality of the imported resources, i.e. their exergetic content, as well as the efficiency of such systems in extracting the exergetic content in their processes [2].

As cities grow in size and complexity, self-organizational behaviours that are typical of open thermodynamic systems emerge. These include diversifying intra-system interactions and increasing their intensity in order to more effectively process the increased energy intake [3]. Such behaviours provide the capacity for growth and expansion prompting cities to seek and destroy more exergetic content. Given the limited nature of resource supplies, this is typically better achieved by prolonging the chain of transformation processes within systems, i.e. keeping materials in systems for longer, which in turn increases their capacity in retaining and circulating quality resources through these longer chains [4,5]. For complex dissipative thermodynamic systems like cities, measuring this gradient of exergy destruction in the overall system serves as a performance indicator of the systems' capability to use resources they import and re-circulate [6]. This very quality is echoed in and underpins the working principles of circular economy where reduce, maintain, re-use, refurbish and recycle are incorporated to address the over-extraction of resources through maximizing transformation efficiencies (reduce), keeping resource in use (maintain) and prolonging next-use chains (re-use, refurbish and recycle) through various strategies. Applying circular-economic principles not only relieves the demand on resource use, but also promotes system adaptations and product design for cutting waste and carbon emissions [7]. The essence of the circular economy is to retain the usefulness of resources through a hierarchy of strategies where higher levels of resource management such as maintenance to extend the current life are preferred, with reuse for a second life, and then repair and remanufacturing further down the hierarchy, before finally recycling to extend the lifespan and number of life cycles of resources [8]. Implementing circular-economic approaches in cities and adapting circularity strategies in industrial processes is thought to ultimately lead to an economically desirable and environmentally sustainable development plan [9].

In cities, these interconnected processes, especially those of energetic and resource flows in between industrial sectors, are perceived as governed by a series of complex interactions and dynamics [10]. As such, ecologically based analogies and network-related approaches, from the explicit study of network topology and structure [11,12] to material flow analysis (MFA) [13] and ecological network analysis (ENA) [14], are very often applied in measuring and quantifying urban energy/material systems.[1] For instance, MFA has been commonly used in tracking and mapping resource flows and stocks entering and leaving a system to help understand the distribution of resources [15]. Similarly, ENA uses inter-sectoral input–output exchanges to investigate the resource flow dynamics between different components of urban systems analogous to those in ecological systems [16,17].[2] Furthermore, the development of open system thermodynamic approaches have also seen the combination of these methods with exergy analysis in quantifying the effects of organizational behaviours of resource systems in cities on their ability to use resources most efficiently [4]. These network analytic formulations of urban resource systems have become some of the most popular tools for investigating the distribution of flows, with an increasing focus on the practical applications in studying resource management in cities through case studies at multi-scale levels [18]. However, unlike the early fundamental works on ecosystem modelling where the analytical boundaries of the models were explored and established [19,20], the current application of these frameworks to urban resource systems is missing such an analytical examination. Since these empirical studies rely on abstractions of cities as networks of resource flows and processes, analytical examinations of these models are required in order to identify behaviours that arise due to the particularities of the models, are external to the specific cities that are studied, and are thus deterministic with respect to the model parameters.

The aim of this work is therefore to provide an analytical assessment of a generic network formulation of these urban resource models that underpins many existing studies, be it explicitly adopted or implicit in their assumptions, that are grounded in open system principles in terms of resource distribution and utilization. In this paper, we present an analytical formulation and examination of a demand-driven network model that accounts for the 'effectiveness of resource utilization' which enables us to better frame the existing empirical efforts and provide a better understanding and prioritization of available policy levers in addressing resource management in cities. The model framework here is adopted from the work by Tan *et al.* [21], which provides a typical exergy-based open system network assessment of

[1]Moving forward, unless otherwise specified, we will use the term 'resource' to refer to the exergetic content and quality of both energetic and material flows.

[2]The use of the phrase 'urban systems' in this work is in reference to systems and processes of material and energetic flows within an urban context and should not be confused with the usage referring to 'systems of cities' within the body of work on urban scaling and complexity.

the sustainability of urban resource use from an ecological-thermodynamic perspective. As we will see in our examination, perceived truisms regarding the effects of increased process efficiency and recycling in managing resource consumption and increased system performance, while intuitive, are the by-products of the network paradigm commonly chosen by the existing empirical studies. In broad terms, our approach provides a network formulation which can specifically connect across applications of material flow and circular-economic analyses.

The rest of this paper is organized as follows. The next section provides a brief overview of the model and our formal analytical specification of it. This is followed by §3 where we present analytically derived expectations of urban performance effectiveness limits as a function of the urban energetic/resource networks topological and performance properties. In §4, we then explore the analytical distribution of the achievable performance levels across various urban structures complemented with a series of Monte Carlo numerical experiments. Section 5 follows up with an overall discussion of the results presented and their implications within different urban domains with specific examples pertaining to circular-economic approaches. This is followed by a brief conclusion and summary of findings.

# 2. A network model of urban flows

As previously mentioned, we use the urban network framework developed by Tan *et al.* [21]. This model and the broader family of which it is a member attempt to provide a benchmark for how well the resources available to cities are used [22]. This is often formulated as a system-wide performance metric expressed as the ratio of the portion of resources beneficially consumed to the total resources externally imported and/or newly extracted by the city. This is then seen as cities' effectiveness in using the inward extra-boundary flows. It could be argued that defining performance metrics solely with respect to the imported/extracted flows is limiting. However, such a formulation does provide a broad and portable definition that can in fact codify efficiency in resource performance in a variety of networks in an urban context. In maintaining consistency of language with Tan *et al.* [21], we will continue to use the term 'effectiveness' in reference to the framework's performance metric. While the metric is in essence one of efficiency, the particular choice of language is to differentiate the metric from the commonly used system-wide efficiency measures within the exergy-focused literature that are calculated based on the ratio of outgoing to incoming extra-boundary resource flows [23,24].

For energy systems, where vertices embody services and processes of a thermodynamic nature, an import-centric metric captures the overall ability of the urban network in extracting full exergetic content from systems' input streams. Similarly, in a material-based network, say that of circular economy where extra-boundary inputs comprise what would otherwise be waste streams, the metric gauges the system's ability in fully recirculating input streams internally. Such a performance formulation provides a versatile indicator compatible for examining both urban resource utilization and circular-economic performance [25,26].

We consider these urban processes and flows as a directed network on $N$ vertices and $E$ edges while allowing disconnected components. The vertices and edges act as stand-ins for urban processes and flows, the nature of which is determined by the domain context. For instance, in a circular-economic network, vertices would denote an industrial/economic sector and the edges the resource flows passed in between [27]. Whereas in an urban energetic network the vertices would represent thermodynamic processes and the edges those of the energetic flows [4]. Figure 1 shows a schematic of such a network where $\Delta_i$ denotes extra-boundary flows into and out of vertex $i$, $X_i^W$ the overall wasted flows at vertex $i$, $X_i^U$ the beneficially used portion of the resource flow, and $F_{ij}$ the flow passed from vertex $i$ to $j$.

Following existing model set-ups [4,14,21,27], we enforce the overall relationship between edges, under flow equilibrium conditions, through the use of two disutility factors $\lambda$ and $\phi$. In our analytical presentation of these models, the parameters $\lambda$ and $\phi$ regulate flow transformation processes taking place inside the vertices. In particular, $\phi$ determines the irrecoverable wasted flow in the process and $\lambda$ regulates the portion of the flow successfully used by the downstream vertex. With reference to figure 1, the model fixes the demand from the downstream vertex, $j$. Vertex $j$ will receive a flow $F_{ij}$ from vertex $i$, meaning that, within $i$, the energy needed to supply $j$ is $F_{ij}/\phi_i\lambda_i$. Of this the useful portion of the total outbound intra-network flow is $F_{ij}((1 - \lambda_i)/\phi_i\lambda_i)$ and the wasted portion is $F_{ij}((1 - \phi_i)/\phi_i)$. The aggregate flow across all the vertices supplied by vertex $i$ is

$$X_i^U = \sum_{j \in S_i}^{N} F_{ij} \frac{1 - \lambda_i}{\phi_i \lambda_i} \tag{2.1a}$$

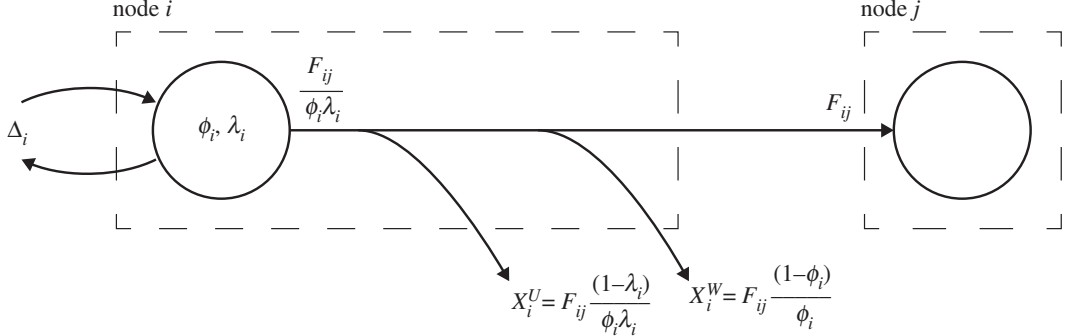

**Figure 1.** Schematic of a sample network illustrating the intra-network in- and out-flows, external imports, exports, wasted and used flows.

and

$$X_i^W = \sum_{j \in S_i}^N F_{ij} \frac{1 - \phi_i}{\phi_i}, \qquad (2.1b)$$

where $S_i$ is the set of vertices supplied by $i$. Hereinafter, we refer to $\phi$ and $\lambda$ as recoverability and conversion disutility factors, respectively. Note that the particular assignment of useful and wasted is context dependent and is further explored in the discussion. Excluding stock accumulation in an equilibrium state, the nodal balance can be written as

$$\Delta_i = \sum_{j, j \neq i}^N F_{ij} + X_i^U + X_i^W - \sum_{j, j \neq i}^N F_{ji} = \sum_{j, j \neq i}^N \frac{F_{ij}}{\phi_i \lambda_i} - F_{ji}, \qquad (2.2)$$

where $\Delta_i$, the resource exchange with the environment needed to balance the vertex, is taken as positive when vertex $i$ is importing from the environment.[3] Note that $\Delta_i$ can be resource flows imported from other cities or those extracted from the city's own surrounding environment. The network's overall effectiveness of resource utilization, $\varepsilon$, can then be formally written as the sum total of all $X_i^U$ divided by the net extra-boundary import

$$\epsilon \underset{=}{\text{def}} \frac{\sum_i^N X_i^U}{\sum_i^N \Delta_i^+}. \qquad (2.3)$$

Note that $\sum_i^N \Delta_i^+$ limits the sum to nodes with a positive $\Delta$ and hence a net import of extra-boundary flows. For the sake of completeness, we formalize this system specification in matrix form. To do so, we define the vectors $\lambda = \{\lambda_1, \lambda_2, \ldots, \lambda_N\}$ and $\phi = \{\phi_1, \phi_2, \ldots, \phi_N\}$ and the diagonal matrices $\Lambda$, $\Phi$ and $\Omega$ as

$$\Lambda = \text{diag}(\lambda), \ \Phi = \text{diag}(\phi), \ \Omega = (\Lambda\Phi)^{-1}. \qquad (2.4)$$

This allows writing equation (2.3) as

$$\epsilon = \frac{\mathbf{1}^T (I - \Lambda) \Omega A \mathbf{1}}{t^T (\Omega A - A^T) \mathbf{1}}, \qquad (2.5)$$

with $t$ a vector assigned as

$$t_i = \begin{cases} 1 & \text{if } \Delta_i > 0, \\ 0 & \text{otherwise,} \end{cases} \qquad (2.6)$$

and $A$ the weighted adjacency such that

$$A = \begin{bmatrix} 0 & \cdots & a_{1N}f_{1N} \\ \vdots & \ddots & \vdots \\ a_{N1}f_{N1} & \cdots & 0 \end{bmatrix}, \qquad (2.7)$$

---

[3]In this work, we consider a temporally static model without stock accumulation. It should be noted that, from a resource utilization perspective, the addition of dynamic stocks only serves to displace system effectiveness estimations in time. This is because stocks that are used by vertices as incoming resources are exergetically indistinguishable from extra-boundary imports and would have required extra-boundary imports in a previous time step in any case.

where, conventionally

$$a_{ij} = \begin{cases} 1 & \text{if there exists a directed edge} \quad \text{from } i \text{ to } j \\ 0 & \text{otherwise.} \end{cases} \tag{2.8}$$

The coefficients, $\lambda_i$ and $\phi_i$, are assumed drawn from a standard uniform distribution on the interval (0, 1), and the flow magnitude, $f_{ij}$, on the interval [0, 1], signifying a lack of particular empirical information regarding system characteristics [4]. In this assignment, the flows are hence normalized against the maximum flow observed in the system. We have intentionally kept the formal specification of these urban models generic and minimal. This enables both portability of the model for analysing various systems of different nature in an urban context and also maximizes the generalization potential of fundamental behaviour inherent to the assumed random nature of network structure and the characterization of disutility factors and flows, which are in general difficult to empirically observe for all urban processes [28].

# 3. Size-independence of the limit to system effectiveness

Having formalized the specification of the model that implicitly underpins empirical studies of urban resource networks, we begin this section by establishing simple limiting boundaries to the system effectiveness as a function of the process disutility factors.

## 3.1. Upper limits to average system-wide resource utilization

**Proposition 3.1.** *For balanced urban networks with identical processes, effectiveness of resource utilization is independent of network size and remains equal to $(1 - \lambda)/(1 - \phi\lambda)$.*

*Identical processes provide that*

$$\forall i, j, k \begin{cases} \lambda_i = \lambda_j = \lambda \\ \phi_i = \phi_j = \phi \\ F_{ij} = F_{jk} = F \end{cases} . \tag{3.1}$$

*And the balanced network assumption provides the minimum requirement such that*

$$\forall i \, |d_i^{\text{in}} = d_i^{\text{out}} = d_i \rightarrow \Delta_i = \sum_j^{d_i} \frac{F_{ij}}{\phi_i \lambda_i} - F_{ji} > 0, \tag{3.2}$$

*where $d_i^{\text{in}}$ and $d_i^{\text{out}}$ denote the in- and out-degree of vertex i, respectively. Coupled with the assumption of identical processes, this yields*

$$\Delta_i = \sum_j^{d_i} F\left(\frac{1}{\phi\lambda} - 1\right) = d_i F\left(\frac{1}{\phi\lambda} - 1\right) > 0. \tag{3.3}$$

*Substituting values back in equations (2 .1a) and (2.2) gives us $X_i^U$, $\Delta_i^+$ and $\varepsilon$ as*

$$X_i^U = \sum_{j,j \neq i}^{d_i} F_{ij} \frac{1 - \lambda_i}{\phi_i \lambda_i} = \sum_{j,j \neq i}^{d_i} F \frac{1 - \lambda}{\phi\lambda} = d_i F \frac{1 - \lambda}{\phi\lambda}, \tag{3.4a}$$

$$\Delta_i^+ = \Delta_i = d_i\left(\frac{1}{\phi\lambda} - 1\right)F \tag{3.4b}$$

and

$$\epsilon = \frac{\sum_i^N X_i^U}{\sum_i^N \Delta_i^+} = \frac{d_i F((1 - \lambda)/\phi\lambda)}{d_i(1/\phi\lambda - 1)F} = \frac{1 - \lambda}{1 - \phi\lambda}. \tag{3.5}$$

It is trivial to show that $\epsilon = (1 - \lambda)/(1 - \phi\lambda)$ is in fact the upper limit to the mean system-wide effectiveness of utilization for given system-wide mean of $\lambda$ and $\phi$. Any edges that are added between the existing vertices of a balanced network would not alter overall system effectiveness. This is because the additional $F((1 - \lambda)/\phi\lambda)$ in the numerator is accompanied by a reduction of net imports of magnitude $F$ at the receiving vertex and an increase of magnitude $F/\phi\lambda$ at the vertex of origin in the denominator. As such, the total count of the summed flows, while increased, remains constant in both the numerator and denominator resulting in the same overall effectiveness. Figure 2 provides a

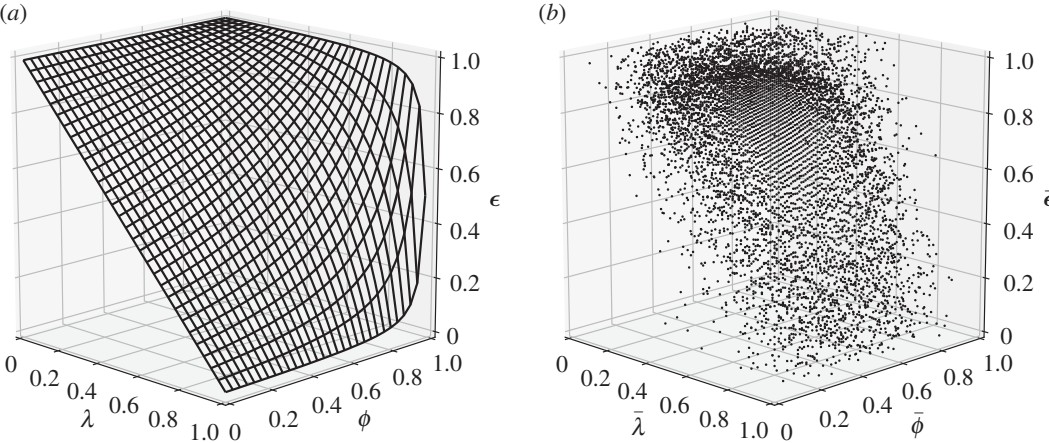

**Figure 2.** Upper-limit boundary of effectiveness of resource utilization, $\epsilon = (1 - \lambda)/(1 - \lambda\phi)$, for urban networks (*a*) and 50 k Monte Carlo estimates of mean effectiveness against network mean $\lambda$ and $\phi$ (*b*)—note that in the absence of conversion disutilities ($\lambda \to 0$) or when there is no waste ($\phi \to 1$), the network only imports as much resources as it has need for by either fully using resources at the point of origin or through eventual recycling of all flows from within the system.

visualization of this effectiveness boundary both analytically and numerically investigated for a series of 50 k Monte Carlo simulations networks of 2–30 vertices with flow and vertex characteristics sampled from a uniform distribution (see electronic supplementary material, appendix for details of the numerical simulations and the direct validation of proposition 3.1). It is crucial to note in figure 2 that the simplified upper limit derived for the case of systems without process-level heterogeneity, panel (*a*), does in fact hold as the limiting envelop for the average behaviour of heterogeneous systems where flows and disutility factors are independent and identically uniformly distributed, panel (*b*).

Although this limit may appear trivial, its significance is twofold. First is that in the absence of flow heterogeneity the upper bound of a system's performance is not a function of the system size, i.e. number of vertices, and is simply determined by average disutility characteristics of the system. Secondly, considering the gradients of $\varepsilon$ with respect to $\lambda$ and $\phi$, it is clear that effectiveness response is more sensitive to changes in average conversion disutility, $\lambda$, than it is to recoverability disutility, $\phi$, with this even more pronounced for systems with heterogeneous vertex efficiencies (figure 2*b*).

## 3.2. Lower limits to average system-wide resource utilization

In addition to the upper limit in proposition 3.1, we also provide the lower limit to effectiveness of resource utilization by considering urban networks comprising solely sinks and sources.

**Proposition 3.2.** *For urban networks comprising solely sources and sinks with identical processes, network effectiveness is independent of network size and equal to $1 - \lambda$.*

*We write the sink or source assumption as*

$$\left.\begin{array}{l} d_i^{\mathrm{in}} = 0 \quad \text{if source} \\ d_i^{\mathrm{out}} = 0 \quad \text{if sink,} \end{array}\right\} \tag{3.6}$$

*and, substituting values back in equation (2.2), we get*

$$\left.\begin{array}{l} \forall i \in \text{sources: } \Delta_i = \sum_{j \in S_i}^{d_i^{\mathrm{out}}} \dfrac{F_{ij}}{\phi_i \lambda_i} - \sum_j^{d_i^{\mathrm{in}}} F_{ji} = \sum_{j \in S_i}^{d_i^{\mathrm{out}}} \dfrac{F}{\phi\lambda} > 0 \to \Delta_i^+ = d_i^{\mathrm{out}} \dfrac{F}{\phi\lambda} \\[3ex] \forall i \in \text{sinks: } \Delta_i = \sum_{j \in S_i}^{d_i^{\mathrm{out}}} \dfrac{F_{ij}}{\phi_i \lambda_i} - \sum_j^{d_i^{\mathrm{in}}} F_{ji} = -\sum_j^{d_i^{\mathrm{in}}} F_{ji} < 0 \to \Delta_i^+ = 0. \end{array}\right\} \tag{3.7}$$

*Finally, substituting for effectiveness results in*

$$\epsilon = \frac{\sum_i^N X_i^U}{\sum_i^N \Delta_i^+} = \frac{((1 - \lambda)/\phi\lambda) \sum_i^{\text{sources}} d_i^{\mathrm{out}} F}{(1/\phi\lambda) \sum_i^{\text{sources}} d_i^{\mathrm{out}} F} = 1 - \lambda. \tag{3.8}$$

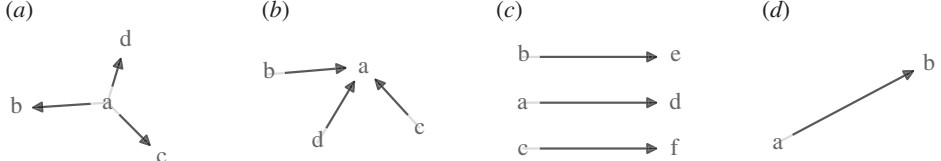

**Figure 3.** Examples of interchangeable networks of varying vertex and edge count with identical effectiveness limit, $\epsilon = 1 - \lambda$, in the absence of heterogeneity of characteristics.

To confirm $\epsilon = 1 - \lambda$ is in fact the lower limit, we follow a reasoning similar to that of the upper limit. Given a source/sink system, the elimination of each existing edge is equivalent to a unit reduction in the degree count of the corresponding source vertex, $d_i$. It can be seen from equation (3.8) that this change in $d_i$ only affects the summation count in both numerator and denominator without a change in the overall ratio. Since this edge elimination does not affect the source/sink-only assumption, the elimination can be continued until the system is reduced to the basic two-vertex single-edge process with the same effectiveness. Figure 3 illustrates this equivalency of systems for a number of networks of different vertex and edge counts. It is beneficial to note that we explicitly do not assume the urban networks to necessarily be connected.

# 4. Order statistics and distribution of effectiveness with network size

By assuming identical vertex characteristics, the previous section provides limits on the ability of urban networks in achieving performance effectiveness given flow homogeneity and average expected disutility characteristics. Here, we consider the overall distribution space of network effectiveness upper limit.

Obtaining a distribution for the effectiveness upper limit requires allowing for the vertex heterogeneity while considering fully connected networks of varying vertex count. If network flows and vertex disutility factors, $\lambda$ and $\phi$, are sampled independently from a given distribution, the heterogeneity of the beneficially used, $X^U$, and net import, $\Delta^+$, across vertices can be described via the distribution of the order statistics of the original distribution. We reiterate that the choice of standard uniform is ultimately arbitrary and only to capture and formalize a lack of information on effectiveness and disutility of individual urban processes. It should, however, be noted that the distributions of max-normalized flows and disutility characteristics have necessarily bounded intervals.

In brief, the order statistics of a set, $S$ of the size $N$, sampled from a distribution comprises the ordered set of the sample elements

$$\{x \in S | x_{(1)} \leq x_{(2)} \leq \cdots \leq x_{(N)}\}, \tag{4.1}$$

where the individual distribution of each order statistic, $x_{(i)}$ from $S$, then depends on the sample size, $N$, and the distribution from which $S$ has been sampled [29]. In the case of our urban network, this means that for a given count of vertices, averages can be derived for $\lambda$ and $\phi$ on each vertex treating $\lambda_i$ and $\phi_i$ as the $i$th-order statistics of the set of vertices. For a uniform distribution, order statistics of a sample can be shown to be Beta distributed [29] enabling us to write

$$\lambda \sim U(0,1) \rightarrow \lambda_{(i)} \sim B(i, N+1-i) \rightarrow \bar{\lambda}_{(i)} = \frac{i}{N+1}, \tag{4.2}$$

where $\bar{\lambda}_{(i)}$ is the mean value of the $i$th-order statistic.

As for the exact distribution of effectiveness when all three parameters, $F$, $\lambda$ and $\phi$, are randomly sampled, analytical tractability becomes an issue. Although exact density functions have been derived for the sum, product and ratios of independent Beta distributions, these are often concerned with only two random variables and in general involve hypergeometric functions [30–32]. Sacrificing absolute accuracy, the probability density of a combination of multiple Beta distributions can be approximated as remaining Beta distributed [30]. This means that we can expect

$$\epsilon \sim B(\alpha, \beta), \text{ where } \begin{cases} \alpha \propto f(N) \\ \text{and } \beta \propto g(N) \end{cases}, \tag{4.3}$$

where $\alpha$ and $\beta$ are the shape factors of the Beta distribution and $f(N)$ and $g(N)$ functions of the system's network size to be determined. Figure 4 shows the least-square estimates of $\alpha$ and $\beta$ fitted to the Monte Carlo simulations of fully connected networks of 2–30 vertices.

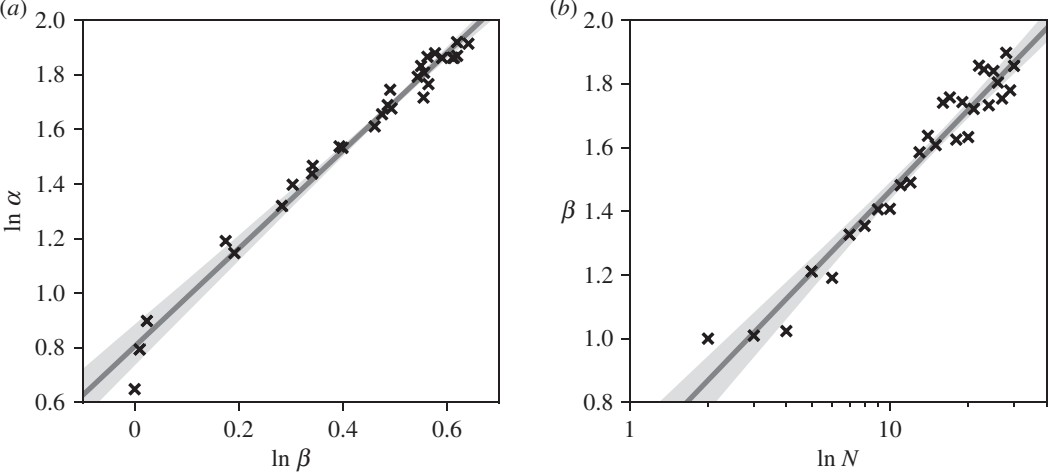

**Figure 4.** Variations of the fitted Beta shape factors, $\alpha$ and $\beta$, against each other and vertex count, $N$, of fully connected networks of up to 30 vertices.

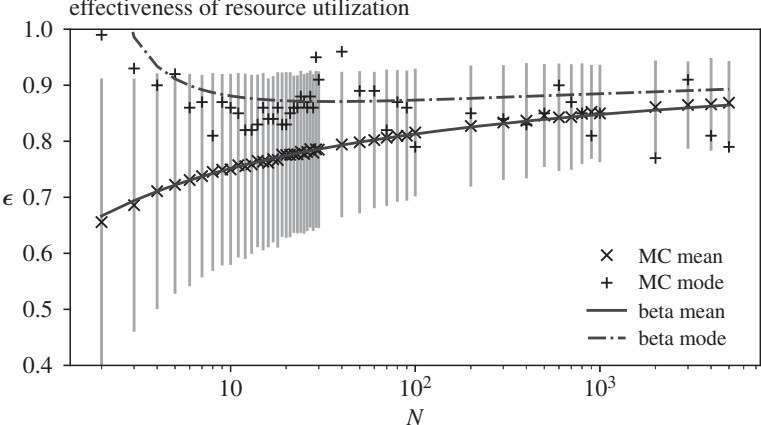

**Figure 5.** Analytical mean and mode of the Beta distribution with shape factors estimated after equation (4.4) overlaid with 50 k Monte Carlo estimates of $\epsilon$ and their standard deviation—note that wider spread in the Monte Carlo mode values are an artefact of the binning $\epsilon$ values to 2 decimal places to obtain the mode.

From these we can approximate $f(N)$, $g(N)$, and hence the analytical mean of the upper limit boundary for the effectiveness as

$$\epsilon \sim B(\alpha, \beta), \text{ where } \begin{cases} \alpha = a\beta^b, \text{ and} \\ \beta = m\ln N + c \end{cases} \rightarrow \bar{\epsilon} = \frac{\alpha}{\alpha + \beta} = \frac{a(m\ln N + c)^b}{a(m\ln N + c)^b + m\ln N + c}, \quad (4.4)$$

where $\bar{\epsilon}$ is the analytical mean of a Beta distributed effectiveness of resource utilization with $a$, $b$, $m$ and $c$ coefficients to be estimated.

Finally, it is important to note that although the limit of equation (4.4) as the vertex count tends to infinity approaches unity, in agreement with proposition 3.1, the gradient of the mean effectiveness with respect to system size also approaches zero. This means that while larger networks are theoretically more effective, on average, with a chance at full effectiveness, in the presence of utility heterogeneities, urban systems of a realistic size are more likely to exhibit effectiveness of resource utilization below 80%. Figure 5 shows the Monte Carlo estimates of the fully connected effectiveness for vertex counts of up to 5000 overlaid with the Beta's mean, estimated using the fit parameters obtained only using networks of up to 30 vertices from figure 4 (see electronic supplementary material, appendix for distribution of $\varepsilon$ as a function of both vertex and edge count). What is clear from the figure is that, although mean effectiveness appears to rise with vertex count accompanied with a decreasing standard deviation, the mode of the effectiveness distribution remains virtually stationary hinting at a practical maximum expectation for a system's effectiveness of resource utilization.

# 5. Discussion and conclusion

We begin the discussion here with a recap of the analytical aspect of the network abstraction of systems in an urban environment before providing an illustration of their potential policy implications specifically as they relate to a circular-economic context. As previously mentioned, the existing body of literature concerned with urban resource systems and circular-economic potential has mostly focused on the application of such network abstraction of urban systems largely for accounting and diagnostic purposes [22]. The analytical formulation in this work makes explicit a number of system behaviours that while intuitive within the context of such systems are ultimately by-products of the implicit network abstraction underlying the existing empirical studies.

## 5.1. System size and resource effectiveness

Here, we have demonstrated analytically that the average behaviour and performance of such urban network abstractions are deterministic and more strongly a function of the distribution of flows and process efficiencies rather than network topology. It is, however, important to note that these average tendencies demonstrated for urban systems correspond to the set of all possible but not plausible system configuration. If we had reason to believe that urban resource systems in their topology are constrained to a particular set of networks, say those that exhibit scale-free properties, then the much narrower selection criteria within the performance distribution of all random graphs would lead us to observe different patterns of dependency on network size and structure for the mean of the utilization effectiveness.

We should also clarify that the deterministic behaviour discussed is of the system's resource performance and not the organization and evolution of systems in cities which are complex and dynamic processes themselves. This is because the urban model examined in this work is concerned with the structure of a system rather than the mechanism by which that structure may have come to be or evolve. To capture these, future work would need to look at formulating these complex dynamics in the network, e.g. through game theoretic approaches for analytical purposes, such that it takes account of organization and sectoral interactions between vertices beyond their pure intensity of resource flow. Going back to the current analytical formulation, one might argue that the specific choice of a uniform distribution is arbitrary and could be considered as guiding the conclusions in the analytical approximation and numerical results shown. We, however, strongly note that although the overall Beta approximation of system effectiveness against its size is specifically tied to the choice of uniform distribution through its order statistics, the upper and lower limit boundaries derived in propositions 3.1 and 3.2 are not dependant on this choice, *per se*. In fact, given the necessarily bounded nature of the distribution of disutility factors and the max-normalized flows, the particularity of how flow and process disutilities might be distributed over their respective domains would only serve to influence the variance around the expected value for resource utilization. The average behaviour of the network with regard to the system-wide expected values of process disutilities, as seen in figures 2 and 5, are thus not significantly affected by the particular choice of the distribution.

The final point to address here is the effect of system size, i.e. the number of sectors/vertices, on the possibility of achieving higher resource effectiveness. We have shown that once process efficiencies are homogeneous across sectors, the system's ability to fully use its inputs are not a function of its size and diversity of its sectors. Coupled with the results in figure 2 for the effectiveness of heterogeneous systems versus their size, it is easy to see that resource utilization in itself does not enjoy considerable size efficiencies. What systems with higher numbers of sectors of more diverse flows would offer is increased chances of finding matching demand and supply of resource flows between existing sectors in the system.

## 5.2. Practical implications and an example

In addressing the portability of the model and its implications, we consider the interpretation of the model as applied to urban resource performance within a circular-economic framework. The model presented here focuses on capturing the impact of multiple life cycles and preservation of resource value, it does not explore the benefits of extending the first life cycle. In this context, each vertex would, for example, represent various industrial activities with edges denoting the resource flows for reuse, refurbish and recycling purposes. Within this mapping of an urban resource system, the

recoverability factor, $\phi$, is then the disutility factor determining the portion of resource flow that can be reclaimed and the conversion disutility factor, $\lambda$, what determines the resource quality through the transformation processes. As an illustration, consider the reuse and recycling of reclaimed bricks. Direct reuse of a brick as another brick corresponds to a low-disutility of conversion, $\lambda \ll 1$. It is perhaps worth noting, however, that unless the overall resource input required to reclaim a material successfully at its original quality is significantly lower than the amount required in its virgin production, conversion disutility $\lambda$ will never truly approach or equal zero [33]. Meanwhile, for a brick crushed and downcycled, conversion disutility increases, $\lambda \gg 0$, as recoverability factor decreases, $\phi \ll 1$, due to higher chances of loss of material in retrieval, transport, and use of the crushed brick. Similar analogies can be made using products from other economic and industrial sectors. We can alternatively consider the potential for cyclic flows involved in the manufacturing, reusing, and recycling of products such as glass bottles and aluminium cans. A low-disutility of conversion, similar to bricks, highlights practices common with glass bottles where regular deliveries of products such as milk rely on collection and reuse of the same cohort of glass bottles keeping the product in use for longer periods. By contrast, recycling regimes for aluminium cans embody higher conversion disutilities and lower recoverabilities, similar to crushed brick although to a smaller extent, as recycled aluminium needs to be reprocessed for manufacturing new cans or other aluminium products. Such a framing of $\lambda$ and $\phi$ is, therefore, compatible with a large number of slightly varying definitions of circular economy [25], and also implicitly captures the hierarchical nature of flow quality in a circular-economic context, albeit on an average-aggregate basis [34]. Consequently, the system's effectiveness of resource utilization quantifies the portion of the input resources the system has managed to re-use whether through enhancements of the characteristic factors of processes or availability of the of intra-system flows.

The levers available to increase resource utilization effectiveness in such systems are then in fact the intuitive steps of reducing waste and recirculating existing flows regardless of the type of the urban resource system considered. The effectiveness of these approaches is, however, potentially more limited at a system-wide scale than might appear. As the conditions used in deriving the upper limit effectiveness boundary in proposition 3.1 suggests and figure 2 illustrates, first steps in moving the overall system performance towards its upper limit, should focus on increasing the homogeneity of process efficiencies across the existing sectors. In terms of increasing system-wide disutility factors, both the analytical and numerical results in figure 2 highlight the greater impact of conversion disutility as opposed to the recoverability of resource flows. This means that, although waste reduction is essential and desired, increasing the amount of 'waste' flows that remain of original functional use, i.e. the brick recovered and re-used as a brick, is much more effective in increasing utilization performance. That is to say that smaller improvements in $\lambda$ as compared with $\phi$ can deliver the same improvement in the system's overall effectiveness of resource utilization. This is while individual firms are far more likely to minimize their wastes than redesign products for lower resource use [35]. Moreover, the minimum requirement of a balanced network in achieving the upper limit of the effectiveness also implies that while a fully connected and circular economy is ideal in terms of resource up/recycling, given its practical infeasibility, interventions need to place attention on creation and maintenance of resource flow pairs that drive urban systems towards balanced networks topologically in conjunction with improving recoverability and conversion characteristics. Nevertheless, it is worth noting that, in time, with information and inventories characterizing urban metabolic processes becoming increasingly accessible, the method presented here provides a pathway to arrive at macroscopic network characterizations of these complex processes.

Data accessibility. The Python script used for the creation and analysis of the Monte Carlo simulations is available at (https://github.com/cip15ha/randomgraph-resource-utilisation) and has been uploaded as part of the electronic supplementary material. The simulation data used for the figures are available from the authors upon request and data available from the Dryad Digital Repository: https://doi.org/10.5061/dryad.kwh70rz0n [36].

Authors' contributions. H.A. and G.M. conceived the study; G.M., H.A. and L.M.T. undertook the numerical study; G.P., Q.L. and H.A. undertook the analytical study; H.A., D.D.T. and M.M. contributed to the discussion; H.A. assembled the final manuscript. All authors gave final approval for publication and agree to be held accountable for the work performed therein.

Competing interests. We declare we have no competing interests.

Funding. G.P. was supported by EPSRC Engineering Complexity Resilience Network Plus (grant no. EP/N010019/1). G.M. was supported by EPSRC City Observatory Research Platform for Innovation and Analytics (grant no. EP/R013411/1). L.M.T. acknowledges support from the Grantham Centre for Sustainable Futures at the University of Sheffield. Q.L. acknowledges support from Sheffield Urban Flows Observatory.

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
