## [Reviewer comments · Royal Society Open Science]

Review History

RSOS-191343.R0 (Original submission)

Review form: Reviewer 1

Is the manuscript scientifically sound in its present form?

Yes

Are the interpretations and conclusions justified by the results?

Yes

Is the language acceptable?

Yes

Do you have any ethical concerns with this paper?

No

Have you any concerns about statistical analyses in this paper?

No

Recommendation?

Accept as is

Comments to the Author(s)

In the paper under consideration, the authors describe and analyse a novel urban network model. The analysis shows that urban performance is not a function of the system size, but depends on the efficiency of the processes of the city and the ability to maintain the quality of the resources through the transformation processes.

The paper is extremely well written and motivated and, although I am not an expert in the field, the technical formulation and analysis look sound. The structure of the paper is also very clear. Therefore, I am positive towards this work and I think it can be published in the current form.

Review form: Reviewer 2**Is the manuscript scientifically sound in its present form?**

Yes

Are the interpretations and conclusions justified by the results?

No

Is the language acceptable?

Yes

Do you have any ethical concerns with this paper?

No

Have you any concerns about statistical analyses in this paper?

Yes

Recommendation?

Reject

Comments to the Author(s)

Please see report (Appendix A).

Decision letter (RSOS-191343.R0)

26-Nov-2019

Dear Dr Arbabi:

Manuscript ID RSOS-191343 entitled "On the Use of Random Graphs in Analysing Resource Utilisation in Urban Systems" which you submitted to Royal Society Open Science, has been reviewed. The comments from reviewers are included at the bottom of this letter.

In view of the criticisms of the reviewers, the manuscript has been rejected in its current form. However, a new manuscript may be submitted which takes into consideration these comments.

Please note that resubmitting your manuscript does not guarantee eventual acceptance, and that your resubmission will be subject to peer review before a decision is made.

Your resubmitted manuscript should be submitted by 25-May-2020. If you are unable to submit by this date please contact the Editorial Office.

Kind regards,
Lianne Parkhouse
Editorial Coordinator
Royal Society Open Science
openscience@royalsociety.org

on behalf of Professor Matjaz Perc (Associate Editor) and R. Kerry Rowe (Subject Editor)
openscience@royalsociety.org

Reviewers' Comments to Author:

Reviewer: 1
Comments to the Author(s)

In the paper under consideration, the authors describe and analyse a novel urban network model. The analysis shows that urban performance is not a function of the system size, but depends on the efficiency of the processes of the city and the ability to maintain the quality of the resources through the transformation processes.

The paper is extremely well written and motivated and, although I am not an expert in the field, the technical formulation and analysis look sound. The structure of the paper is also very clear. Therefore, I am positive towards this work and I think it can be published in the current form.

Reviewer: 2
Comments to the Author(s)

Please see report (attached).

Author's Response to Decision Letter for (RSOS-191343.R0)

See Appendix B.

RSOS-200087.R0

Review form: Reviewer 3

Is the manuscript scientifically sound in its present form?

Yes

Are the interpretations and conclusions justified by the results?

No

Is the language acceptable?

Yes

Do you have any ethical concerns with this paper?

No

Have you any concerns about statistical analyses in this paper?

No

Recommendation?

Accept with minor revision (please list in comments)

Comments to the Author(s)

In this paper, the authors try to provide an analytical assessment of generic network formulation of urban resource models. The paper presents and analytical formulation and examination of a demand-driven network model that accounts for the effectiveness of resource utilization which enables the authors to better frame the existing empirical efforts and provide a better understanding of available policy levers in addressing resource management in cities.

I think the subject is adequate and has enough to be published with some small changes. These changes are format and not content.

In my opinion, the abstract must be rewritten. It does not reflect well what is done and, it seems like an introduction that does not correspond in the abstract. Technically an abstract is a set of short and organized statements that describe, synthesize and comprehensively represent the main ideas of a scientific work. But in this case, it does not describe what is done comprehensively, nor does it represent the main ideas of the article.

More often than not, the sections of the papers must be:

Introduction (to introduce the subject of study)

Related works (study related papers and motivation)

Methodology (how the objectives will be achieved, usually including a flow chart),

3.1 A network Model of Urban flows

3.2 Size-independence of the Limit to System Effectiveness

3.3 Order Statistics and Distribution of Effectiveness with network Size

Experimental results (case study), discussion and conclusions.

Review form: Reviewer 4

Is the manuscript scientifically sound in its present form?

Yes

Are the interpretations and conclusions justified by the results?

Yes

Is the language acceptable?

Yes

Do you have any ethical concerns with this paper?

No

Have you any concerns about statistical analyses in this paper?

No

Recommendation?

Accept as is

Comments to the Author(s)

The manuscripts exploring very important issues on resource utilization in urban systems. It deals with the fundamental aspects of networks and how structural features matter for flow indicators. The manuscript has a clear structure and presents the work in a clear and transparent way. I think the manuscript should be published. However, I think the paper could benefit if the very last section would be enriched by one or two more examples to illustrate what kind of insights can be expected from such an analysis.

Decision letter (RSOS-200087.R0)

06-Mar-2020

Dear Dr Arbabi

On behalf of the Editor, I am pleased to inform you that your Manuscript RSOS-200087 entitled "On the Use of Random Graphs in Analysing Resource Utilisation in Urban Systems" has been accepted for publication in Royal Society Open Science subject to minor revision in accordance with the referee suggestions. Please find the referees' comments at the end of this email.

The reviewers and Subject Editor have recommended publication, but also suggest some minor revisions to your manuscript. Therefore, I invite you to respond to the comments and revise your manuscript.

- **Ethics statement**

- **Data accessibility**

It is a condition of publication that all supporting data are made available either as supplementary information or preferably in a suitable permanent repository. The data

accessibility section should state where the article's supporting data can be accessed. This section should also include details, where possible of where to access other relevant research materials such as statistical tools, protocols, software etc can be accessed. If the data has been deposited in an external repository this section should list the database, accession number and link to the DOI for all data from the article that has been made publicly available. Data sets that have been deposited in an external repository and have a DOI should also be appropriately cited in the manuscript and included in the reference list.

If you wish to submit your supporting data or code to Dryad (<http://datadryad.org/>), or modify your current submission to dryad, please use the following link:
<http://datadryad.org/submit?journalID=RSOS&manu=RSOS-200087>

- **Competing interests**

- **Authors' contributions**

- **Acknowledgements**

- **Funding statement**

Because the schedule for publication is very tight, it is a condition of publication that you submit the revised version of your manuscript before 15-Mar-2020. Please note that the revision deadline will expire at 00.00am on this date. If you do not think you will be able to meet this date please let me know immediately.

on behalf of Professor Matjaz Perc (Associate Editor) and R. Kerry Rowe (Subject Editor)
openscience@royalsociety.org

Reviewer comments to Author:
Reviewer: 3

Comments to the Author(s)

In this paper, the authors try to provide an analytical assessment of generic network formulation of urban resource models. The paper presents and analytical formulation and examination of a demand-driven network model that accounts for the effectiveness of resource utilization which enables the authors to better frame the existing empirical efforts and provide a better understanding of available policy levers in addressing resource management in cities.

I think the subject is adequate and has enough to be published with some small changes. These changes are format and not content.

In my opinion, the abstract must be rewritten. It does not reflect well what is done and, it seems like an introduction that does not correspond in the abstract. Technically an abstract is a set of short and organized statements that describe, synthesize and comprehensively represent the main ideas of a scientific work. But in this case, it does not describe what is done comprehensively, nor does it represent the main ideas of the article.

More often than not, the sections of the papers must be:

Introduction (to introduce the subject of study)

Related works (study related papers and motivation)

Methodology (how the objectives will be achieved, usually including a flow chart),

3.1 A network Model of Urban flows

3.2 Size-independence of the Limit to System Effectiveness

3.3 Order Statistics and Distribution of Effectiveness with network Size

Experimental results (case study), discussion and conclusions.

Reviewer: 4

Comments to the Author(s)

The manuscripts exploring very important issues on resource utilization in urban systems. It deals with the fundamental aspects of networks and how structural features matter for flow indicators. The manuscript has a clear structure and presents the work in a clear and transparent way. I think the manuscript should be published. However, I think the paper could benefit if the very last section would be enriched by one or two more examples to illustrate what kind of insights can be expected from such an analysis.

Author's Response to Decision Letter for (RSOS-200087.R0)

See Appendix C.

Decision letter (RSOS-200087.R1)

19-Mar-2020

Dear Dr Arbabi,

It is a pleasure to accept your manuscript entitled "On the Use of Random Graphs in Analysing Resource Utilisation in Urban Systems" in its current form for publication in Royal Society Open Science. The comments of the reviewer(s) who reviewed your manuscript are included at the foot of this letter.

Please ensure that you send to the editorial office an editable version of your accepted manuscript, and individual files for each figure and table included in your manuscript. You can

send these in a zip folder if more convenient. Failure to provide these files may delay the processing of your proof. You may disregard this request if you have already provided these files to the editorial office.

on behalf of Professor Matjaz Perc (Associate Editor) and R. Kerry Rowe (Subject Editor)
openscience@royalsociety.org

Appendix A

Review of RSOS-191343

General Comments

The paper uses a network approach to represent resource use in urban systems and finds that if you make some simplifying assumptions, the performance is not a function of system size, rather by average efficiency characteristics of the processes in the city.

Unfortunately, I find the assumptions are too simplifying and guide the conclusion rather more than any valid representation of urban systems. ie the model and the assumptions together do not represent urban systems well and the result is more about random graphs than cities.

Among the assumptions the authors ignore net additions, assume vertices (processes) are identical, in degree and out degree are the same (balanced), and any heterogeneity in flows is taken from a standard uniform distribution. On Page 10 at line 9 the authors themselves acknowledge that the choice of a uniform distribution is pivotal in the approximation of numerical results.

The oversimplification of urban systems fails to acknowledge the specialised, heterogeneous nature of economic and productive sectors that use different resources, and threshold effects that come with size and clustering of resource-use activities.

There is some confusion between 'efficiency' and 'efficacy'. Efficacy is the *ability* to produce a desired or intended result. In complex systems, it may be interpreted to be about whether or not the system retains function or achieves some purpose. Efficiency is a measure of the extent to which input is well used for an intended task or function (e.g. to maintain efficacy) and I believe the author's have confounded the two topics in their definition of effectiveness (page 4 lines 24-27)

Other than those lines abovementioned I cannot see the argument why Equation 3 is about effectiveness except that the authors define it as such. It looks much more like an efficiency measure.

In general I am unsure of the consistent use of efficiency and effectiveness terminology and e.g. the authors sometimes use "efficiency" to refer to parameters (λ , ϕ) that they also refer to as 'disutility factors'

The introduction, analysis and discussion are not strongly connected to urban systems which is really only mentioned in the example of recycling of bricks. It is arguably false to say that urban systems can be represented with random graphs. It has been shown empirically that there are macroscopic returns to scale (sub-linear power law relations) in energy consumption {Bettencourt, 2007 #155} and, from a microscopic analysis, in economic productivity (Lobo et al. PLoS ONE 2013).

The methodology derives explicitly from a forthcoming article, reference [22] and it is uncertain whether that has been successfully peer-reviewed. Given that the results from the method are so strongly coupled with the validity of that research I would suggest rejection until the predecessor of this paper is accepted and published. Otherwise there is intransparency on the derivation of the method is or whether the present paper is an advance on the currently inaccessible reference [22].

The present paper *does* present a useful network formalism that connects to concepts of material flow analysis MFA and on page 10 (Line 38-52) the example of recycling/reclaiming bricks connects usefully to topics of circular economy (more so than urban systems in general). I think that the method presented could be used with process data from MFA or life-cycle inventories to arrive at macroscopic network efficiency results on complex production processes.

Specific Comments

Page 2 the elaboration on “Overshoot Day”, Figure 1 and the first reference are unnecessary and

Page 8 line 56 I assume densities refers to probability density?

Page 3 line 29 the terminology on “resource use” can be put in a footnote, moving the words ...”moving forward and unless otherwise specified we will use the term resource use to refer to the exergetic content and quality of both energetic and material flows.”

References

Lobo J, Bettencourt LMA, Strumsky D, West GB (2013) Urban Scaling and the Production Function for Cities. PLoS ONE 8(3): e58407.

<https://doi.org/10.1371/journal.pone.0058407>

Appendix B

January 16, 2020

Dear Editors,

We wish to submit our revised research manuscript titled '*On the Use of Random Graphs in Analysing Resources Utilisation in Urban Systems*' to be re-considered for publication in the Journal of Royal Society Open Science having incorporated and responded to the revisions suggested and issues raised by the reviewers.

Addressing the main concern of Reviewer 2, we would like to confirm that the referenced manuscript, *An Ecological-Thermodynamic Approach to Urban Metabolism: Measuring Resource Utilization with Open System Network Effectiveness Analysis*,¹ was peer-reviewed and accepted for publication in Applied Energy prior to the submission of the current manuscript and has been available in press since early August.

An itemised summary of other changes made and comments provided in addressing the points raised by Reviewer 2 is appended to the back of this letter. Where appropriate, these changes have been highlighted in different colours in the revised manuscript corresponding to the response to which they relate.

Yours Sincerely,

Hadi Arbabi^{1,*}, Giuliano Punzo², Gregory Meyers¹, Ling Min Tan¹, Qianqian Li¹, Danielle Densley Tingley¹, Martin Mayfield¹

¹ Department of Civil & Structural Engineering, the University of Sheffield, S1 3JD, UK

² Department of Automatic Control & Systems Engineering, the University of Sheffield, S1 3JD, UK

* Correspondence: h.arbabi@sheffield.ac.uk; Tel.: +44 (0) 114 222 5728

Reviewer: 1

“In the paper under consideration, the authors describe and analyse a novel urban network model. The analysis shows that urban performance is not a function of the system size, but depends on the efficiency of the processes of the city and the ability to maintain the quality of the resources through the transformation processes.

The paper is extremely well written and motivated and, although I am not an expert in the field, the technical formulation and analysis look sound. The structure of the paper is also very clear. Therefore, I am positive towards this work and I think it can be published in the current form.”

We thank the reviewer for their positive feedback.

Reviewer: 2

We thank the reviewer for their time and careful examination of our manuscript. We have broken down and rearranged their comments in order to better address the reviewer's concerns thematically and implement their suggestions as appropriate. The colour coding is used to highlight substantial changes made in the manuscript corresponding to each point raised below. The remaining alterations are visible through tracked changes in red typeface.

1. "The methodology derives explicitly from a forthcoming article, reference [22] and it is uncertain whether that has been successfully peer-reviewed. Given that the results from the method are so strongly coupled with the validity of that research I would suggest rejection until the predecessor of this paper is accepted and published. Otherwise there is intransparency on the derivation of the method is or whether the present paper is an advance on the currently inaccessible reference [22]." [REDACTED]

As mentioned in the opening cover letter, we confirm that the referenced manuscript, 'An Ecological-Thermodynamic Approach to Urban Metabolism: Measuring Resource Utilization with Open System Network Effectiveness Analysis',¹ was accepted for publication in *Applied Energy* prior to the submission of the current manuscript but not assigned a doi or an issue. Since early August, it has been available in press and its reference within the current manuscript has now been updated to reflect full citation information. We apologise for the unclear nature of the language used in the references making it difficult to discern whether the article had indeed undergone peer-review successfully.

See page 11 reference 21.

2. "The introduction, analysis and discussion are not strongly connected to urban systems which is really only mentioned in the example of recycling of bricks. It is arguably false to say that urban systems can be represented with random graphs. It has been shown empirically that there are macroscopic returns to scale (sub-linear power law relations) in energy consumption {Bettencourt, 2007 #155} and, from a microscopic analysis, in economic productivity (Lobo et al. PLoS ONE 2013)." [REDACTED]

The reviewer is correct in arguing that it would be highly inaccurate to represent *systems of cities* as random graphs, especially with reference to the urban scaling body of literature. The use of the phrase 'urban systems' in this work, however, is in reference to systems and processes of material and energetic flows within an urban context and should not be confused with the usage referring to 'systems of cities' within the body of work on urban scaling, agglomeration advantage, and complexity. The first mention of urban systems within the text is now accompanied by an endnote highlighting the above semantic usage. Overall references to 'urban systems' throughout the text have also been modified and we hope that these are now less open to misunderstanding.

See endnote 2.

3. "The present paper does present a useful network formalism that connects to concepts of material flow analysis MFA and on page 10 (Line 38-52) the example of recycling/reclaiming bricks connects usefully to topics of circular economy (more so than urban systems in general). I think that the method presented could be used with process data from MFA or life-cycle inventories to arrive at macroscopic network efficiency results on complex production processes." [REDACTED]

We appreciate the reviewer's acknowledgment of the utility of the work to the material flow analysis, circular-economic analysis, and urban metabolism literature and the connection demonstrated with the brick reclamation example. As mentioned above, the manuscript's main preoccupation has in fact been with such system-level characterisation of flow processes in an intra-urban context, i.e. material flow, circular economy, etc. We have now attempted to convey this more explicitly by signposting this both within the introduction and also towards the end of the manuscript.

See page 2 lines 46-47 and page 10 lines 21-23.

4. "Unfortunately, I find the assumptions are too simplifying and guide the conclusion rather more than any valid representation of urban systems. ie the model and the assumptions together do not represent urban systems well and the result is more about random graphs than cities.

The oversimplification of urban systems fails to acknowledge the specialised, heterogeneous nature of economic and productive sectors that use different resources, and threshold effects that come with size and clustering of resource use activities.”

We understand that the main objections of Reviewer 2 to

- (i) the use of random networks and
- (ii) the failure to acknowledge the efficiencies and agglomeration effects associated with clustering and specialisation

are rooted in the assumption that the method has been used for modelling and simulation in a ‘system of cities’ context given the reviewer’s references to the body of work by Bettencourt and colleagues. We agree with the reviewer that random graphs as used in the present work are not representative models of systems of cities. However, as already noted, the application and formulation at the core of this work seek to address resource flow systems within urban environments. The random network formulation and the simplifying assumptions made explicit in this work are, in practice, consistent with the existing literature within the material flow analysis, ecological network analysis, and energy systems domains which implicitly rely on similar formulation in their work.²⁻⁵

5. “Among the assumptions the authors ignore net additions, assume vertices (processes) are identical, in degree and out degree are the same (balanced), and any heterogeneity in flows is taken from a standard uniform distribution. On Page 10 at line 9 the authors themselves acknowledge that the choice of a uniform distribution is pivotal in the approximation of numerical results.”

In addressing the reviewer’s concern regarding the exclusion of stocks, we note that, from a resource utilisation perspective, the addition of dynamic stocks only serves to displace system effectiveness estimations in time. This is because stocks that are used by vertices as incoming resources are energetically indistinguishable from extra-boundary imports and would have required extraction or extra-boundary imports in a previous time step in any case. We have now made this clear in an endnote following the introduction of Equation 2.

See endnote 3.

As for the criticism regarding the compounding simplifying assumptions, we need to be clear that the assumptions enumerated by the reviewer are considered in a more discretised matter and implemented separately to establish specific system behaviours. The assumptions relating to balanced networks in Proposition 1 and sink/source only networks in Proposition 2 have been made to specifically establish the limiting envelop for the system-wide performance and are not meant as generic behavioural assumptions. We understand that this may not been immediately clear in Section 4 which seeks to establish such upper- and lower-limit limits for the system resource utilisation. We have attempted to clarify this by putting each proposition under a relevant sub-heading making clear references to the nature of the boundary being formulated.

See page 5 sub-section ‘Upper limits to average system-wide resource utilisation’ and page 6 sub-section ‘Lower limits to average system-wide resource utilisation’. See also page 6 lines 3-7.

While acknowledging that we have in fact established the *analytical* upper- and lower-limits using independent but system-wide constant values of λ and ϕ which ignores process-level heterogeneities for convenience, the propositions still hold valid against Monte Carlo experiments that do incorporate process heterogeneities as the reviewer has noted. We have made this clear in the paragraph immediately preceding Figure 2 which shows the results of the Proposition 1 and the Monte Carlo experiments.

See page 6 lines 12-15.

The reviewer is additionally concerned that the choice of the uniform distribution as the sole means of accounting for the process heterogeneities guides the model conclusion and is even acknowledged as such by the authors. This is not in fact the case. We acknowledge that better language could have been used to clear any doubts. The portion of manuscript in discussion has been meant as a clarification that although the choice might *appear* pivotal,

it will not significantly affect the average expected value of the utilisation and would only serve to determine the variance of its distribution. We have edited the language in that section so better clarify this.

See page 9 lines 11-21.

6. "There is some confusion between 'efficiency' and 'efficacy'. Efficacy is the ability to produce a desired or intended result. In complex systems, it may be interpreted to be about whether or not the system retains function or achieves some purpose. Efficiency is a measure of the extent to which input is well used for an intended task or function (e.g. to maintain efficacy) and I believe the author's have confounded the two topics in their definition of effectiveness (page 4 lines 24-27)

Other than those lines abovementioned I cannot see the argument why Equation 3 is about effectiveness except that the authors define it as such. It looks much more like an efficiency measure." [REDACTED]

We are inclined to agree with the reviewer's observation that the metric presented is in essence one of efficiency. However, our work builds on the work by Tan et al., and we follow that language for consistency. This work differentiates the metric from the commonly used system-wide efficiency measures within the exergy-based literature that are calculated based on the ratio of outgoing to incoming extra-boundary resource flows rather than those concerned with the *efficiency* of the resource utilisation within the system.^{6,7} This is a subtle but distinct difference, and we thus follow this language set Tan et al. by referring to the metric as one of *effectiveness*.

In order to make this clear and avoid confusion, we have added a similar explanation to that above before the introduction of the model framework.

See page 3 lines 17-21.

7. "In general I am unsure of the consistent use of efficiency and effectiveness terminology and e.g. the authors sometimes use 'efficiency' to refer to parameters (λ , ϕ) that they also refer to as 'disutility factors'"

We apologise for the occasional inconsistency in the language used regarding process coefficients λ and ϕ . These have now been corrected within the manuscript such that they are consistently referred to as 'disutility factors'. The mentions of 'efficiency' with regard to process/flow factors have thus been wholly eliminated.

8. "Page 2 the elaboration on 'Overshoot Day', Figure 1 and the first reference are unnecessary"

As per the recommendation of the reviewer the introductory sentence, figure, and relevant citation have now been edited out.

See page 1.

9. "Page 8 line 56 I assume densities refers to probability density?"

The two instances of 'densities' have now been edited to refer to density functions and probability density as suited.

See page 8 line 2.

10. "Page 3 line 29 the terminology on 'resource use' can be put in a footnote, moving the words ... 'moving forward and unless otherwise specified we will use the term resource use to refer to the exergetic content and quality of both energetic and material flows.'"

While we are unaware of the editorial policy at Open Science regarding the use of footnotes and endnotes, the phrase has currently been incorporated as an endnote as suggested by the reviewer along with a few others. We would incorporate these notes as parenthetical statements within the text should the editorial guidelines require an elimination of endnote material.

See endnote 2.

References

1. Tan, L. M., Arbabi, H., Brockway, P. E., Densley Tingley, D. & Mayfield, M. An ecological-thermodynamic approach to urban metabolism: Measuring resource utilization with open system network effectiveness analysis. *Applied Energy* **254**, 113618 (2019).
2. Tan, L. M. *et al.* Ecological Network Analysis on Intra-City Metabolism of Functional Urban Areas in England and Wales. *Resources, Conservation and Recycling* **138**, 172–182 (2018).
3. Haas, W., Krausmann, F., Wiedenhofer, D. & Heinz, M. How Circular Is the Global Economy?: An Assessment of Material Flows, Waste Production, and Recycling in the European Union and the World in 2005. *Journal of Industrial Ecology* **19**, 765–777 (2015).
4. Zhang, Y., Yang, Z., Fath, B. D. & Li, S. Ecological network analysis of an urban energy metabolic system: Model development, and a case study of four Chinese cities. *Ecological Modelling* **221**, 1865–1879 (2010).
5. Bristow, D. & Kennedy, C. Maximizing the Use of Energy in Cities Using an Open Systems Network Approach. *Ecological Modelling* **250**, 155–164 (2013).
6. Zhang, B. *et al.* Exergy-based systems account of national resource utilization: China 2012. *Resources, Conservation and Recycling* **132**, 324–338 (2018).
7. Rosen, M. A. Evaluation of energy utilization efficiency in Canada using energy and exergy analyses. *Energy* **17**, 339–350 (1992).

Appendix C

March 12, 2020

Dear Editors,

We wish to re-submit our lightly revised research manuscript titled '*On the Use of Random Graphs in Analysing Resources Utilisation in Urban Systems*' for publication in the Journal of Royal Society Open Science having incorporated and responded to reviewers following an accept with minor revisions decision.

An itemised summary of changes made and comments provided in addressing the points raised by Reviewers 3 and 4 is appended to the back of this letter.

Yours Sincerely,

Hadi Arbabi^{1,*}, Giuliano Punzo², Gregory Meyers¹, Ling Min Tan¹, Qianqian Li¹, Danielle Densley Tingley¹, Martin Mayfield¹

¹ Department of Civil & Structural Engineering, the University of Sheffield, S1 3JD, UK

² Department of Automatic Control & Systems Engineering, the University of Sheffield, S1 3JD, UK

* Correspondence: h.arbabi@sheffield.ac.uk; Tel.: +44 (0) 114 222 5728

Reviewer: 3

1. "In my opinion, the abstract must be rewritten. It does not reflect well what is done and, it seems like an introduction that does not correspond in the abstract. Technically an abstract is a set of short and organized statements that describe, synthesize and comprehensively represent the main ideas of a scientific work. But in this case, it does not describe what is done comprehensively, nor does it represent the main ideas of the article."

To address the reviewer's concern, we have incorporated the generic parts of the abstract back into the introduction and have re-written the abstract to more concisely and explicitly describe, synthesise, and represent the main ideas of the manuscript:

"Urban resource models increasingly rely on implicit network formulations. Resource consumption behaviours documented in the existing empirical studies are ultimately by-products of the network abstractions underlying these models. Here we present an analytical formulation and examination of a generic demand-driven network model that accounts for the effectiveness of resource utilisation and its implications for policy levers in addressing resource management in cities. We establish simple limiting boundaries to systems' resource effectiveness. These limits are found not to be a function of system size and to be simply determined by the system's average ability to maintain resource quality through its transformation processes. We also show that resource utilisation in itself does not enjoy considerable size efficiencies with larger and more diverse systems only offering increased chances of finding matching demand and supply between existing sectors in the system."

2. "More often than not, the sections of the papers must be:
 - Introduction (to introduce the subject of study)
 - Related works (study related papers and motivation)
 - Methodology (how the objectives will be achieved, usually including a flow chart),
 - 3.1 A network Model of Urban flows
 - 3.2 Size-independence of the Limit to System Effectiveness
 - 3.3 Order Statistics and Distribution of Effectiveness with network Size
 - Experimental results (case study), discussion and conclusions."

While we appreciate disciplinary differences in structuring sections of a research manuscript, the format suggested by the reviewer does not in fact change the overall structure of the existing manuscript and simply adds another level to the existing headings by combining the three existing first-level headings under an otherwise empty 'Methodology' section. Given this and the comments from the other three reviewers, we have left the existing structure as is.

Reviewer: 4

1. “The manuscripts exploring very important issues on resource utilization in urban systems. It deals with the fundamental aspects of networks and how structural features matter for flow indicators. The manuscript has a clear structure and presents the work in a clear and transparent way. I think the manuscript should be published. However, I think the paper could benefit if the very last section would be enriched by one or two more examples to illustrate what kind of insights can be expected from such an analysis.”

We thank the reviewer for their positive feedback. To provide a broader example set we have added an extension to the existing case of the bricks discussing the generality of the model formulation for a wide range of manufactured products similar to glass bottles and aluminium cans:

“Similar analogies can be made using products from other economic and industrial sectors. We can alternatively consider the potential for cyclic flows involved in the manufacturing, reusing, and recycling of products such as glass bottles and aluminium cans. A low-disutility of conversion, similar to bricks, highlights practices common with glass bottles where regular deliveries of products such as milk rely on collection and reuse of the same cohort of glass bottles keeping the product in use for longer periods. In contrast, recycling regimes for aluminium cans embody higher conversion disutilities and lower recoverabilities, similar to crushed brick although to a smaller extent, as recycled aluminium needs to be reprocessed for manufacturing new cans or other aluminium products.”